# Comparison of L- and D-Amino Acids for Bacterial Imaging in Lung Infection Mouse Model

**DOI:** 10.3390/ijms23052467

**Published:** 2022-02-23

**Authors:** Yuka Muranaka, Asuka Mizutani, Masato Kobayashi, Koya Nakamoto, Miki Matsue, Kodai Nishi, Kana Yamazaki, Ryuichi Nishii, Naoto Shikano, Shigefumi Okamoto, Keiichi Kawai

**Affiliations:** 1Division of Health Sciences, Graduate School of Medical Sciences, Kanazawa University, 5-11-80 Kodatsuno, Kanazawa 920-1192, Ishikawa, Japan; yukarisa93@stu.kanazawa-u.ac.jp (Y.M.); kou.nakamoto@sage.ocn.ne.jp (K.N.); 2Faculty of Health Sciences, Institute of Medical, Pharmaceutical and Health Sciences, Kanazawa University, 5-11-80 Kodatsuno, Kanazawa 920-1192, Ishikawa, Japan; mizutani.a@staff.kanazawa-u.ac.jp (A.M.); kobayasi@mhs.mp.kanazawa-u.ac.jp (M.K.); sokamoto@mhs.mp.kanazawa-u.ac.jp (S.O.); 3Ishikawa Prefectural Institute of Public Health and Environmental Science, 1-11, Taiyogaoka, Kanazawa 920-1154, Ishikawa, Japan; mmiikkii_0804@pref.ishikawa.lg.jp; 4Department of Radioisotope Medicine, Atomic Bomb Disease Institute, Nagasaki University, 1-12-4 Sakamoto, Nagasaki 852-8523, Nagasaki, Japan; koudai@nagasaki-u.ac.jp; 5Department of Molecular Imaging and Theranostics, Institute for Quantum Medical Science, Quantum Life and Medical Science Directorate, National Institutes for Quantum Science and Technology, 4-9-1 Anagawa, Inage, Chiba 263-8555, Chiba, Japan; yamazaki.kana@qst.go.jp (K.Y.); nishii.ryuichi@qst.go.jp (R.N.); 6Department of Radiological Sciences, Ibaraki Prefectural University of Health Sciences, 4669-2 Ami, Inashiki 300-0394, Ibaraki, Japan; sikano@ipu.ac.jp; 7Advanced Health Care Science Research Unit, Innovative Integrated Bio-Research Core Institute for Frontier Science Initiative, Kanazawa University, 5-11-80 Kodatsuno, Kanazawa 920-0942, Ishikawa, Japan; 8Biomedical Imaging Research Center, University of Fukui, 23-3 Matsuokashimoaizuki, Eiheiji 910-1193, Fukui, Japan

**Keywords:** bacterial imaging, nuclear medicine imaging, bacterial infection, amino acids, methionine

## Abstract

The effectiveness of L- and D-amino acids for detecting the early stage of infection in bacterial imaging was compared. We evaluated the accumulation of ^3^H-L-methionine (Met), ^3^H-D-Met, ^3^H-L-alanine (Ala), and ^3^H-D-Ala in *E. coli* EC-14 and HaCaT cells. Biological distribution was assessed in control and lung-infection-model mice with EC-14 using ^3^H-L- and D-Met, and ^18^F-FDG. A maximum accumulation of ^3^H-L- and D-Met, and ^3^H-L- and D-Ala occurred in the growth phase of EC-14 in vitro. The accumulation of ^3^H-L-Met and L-Ala was greater than that of ^3^H-D-Met and D-Ala in both EC-14 and HaCaT cells. For all radiotracers, the accumulation was greater in EC-14 than in HaCaT cells at early time points. The accumulation was identified at 5 min after injection in EC-14, whereas the accumulation gradually increased in HaCaT cells over time. There was little difference in biodistribution between ^3^H-L-and D-Met except in the brain. ^3^H-L- and D-Met were sensitive for detecting areas of infection after the spread of bacteria throughout the body, whereas ^18^F-FDG mainly detected primary infection areas. Therefore, ^11^C-L- and D-Met, radioisotopes that differ only in terms of ^3^H labeling, could be superior to ^18^F-FDG for detecting bacterial infection in lung-infection-model mice.

## 1. Introduction

Bacterial infections remain a threat to human life, despite the advancement of medical technology. In recent years, the appearance of drug-resistant strains of various pathogens including methicillin-resistant *Staphylococcus aureus* and fluoroquinolone-resistant *Escherichia coli* [1,2] has heightened the sense of urgency regarding treatment of infectious diseases. Bacterial infections are caused by infection of the host with pathogenic bacteria, and the pathogenesis varies depending on the type of causative bacteria [3]. If there is a dysregulated host response, the bacterial infection can lead to severe sepsis, organ dysfunction, and ultimately death [4,5].

Reducing the risk associated with infection by these bacteria requires a diagnostic method able to detect a rapid increase in the growth activity of pathogenic bacteria. Bacterial culture is the main method currently used to identify the bacterial type and thus enable establishment of an effective treatment strategy [6]. Polymerase chain reaction (PCR) and other methods are also clinically available [7]; however, these are invasive because they require specimens to be collected, and there is a delay before the results are available.

Nuclear medicine imaging is a noninvasive diagnostic method for whole-body scanning in a short time. Pathogen-specific imaging agents target the metabolism of carbohydrates, bacterial folate biosynthesis, iron transport systems, components of and selective binders to the bacterial cell wall, substrates for intracellular bacterial proteins, and antimicrobial peptides as specific bacterial pathways [8,9]. In the metabolism of carbohydrates and components of the bacterial cell wall, 2-deoxy-2-[^18^F]fluoro-D-glucose (^18^F-FDG) and [S-methyl-^11^C]-L-methionine (^11^C-L-Met), radiotracers in clinical use for tumor imaging [10], have already been applied for bacterial imaging [11,12]. These nutrients include carbon sources (D-glucose), nitrogen sources (amino acids), and minerals that are essential requirements for bacterial growth [13]. Amino acids of organic nitrogen sources in particular are a major source of nutrients for bacteria and components of cell walls. Therefore, we consider that amino acid imaging would enable the detection of increased growth activity of pathogenic bacteria. Among amino acids, L-amino acids are incorporated as nutrients in human cells, whereas D-amino acids are rarely used in human cells but are used as cell wall components in bacteria [14]. The aim of this study was to compare the suitability of L- and D-amino acids for bacterial nuclear medicine imaging to detect the early stage of infection. As we focused on L-methionine (L-Met) and D-Met, an optical isomer of L-Met, and α-[N-methyl-^11^C]-methylaminoisobutyric acid ([^11^C]MeAIB), a chemical analog of alanine (Ala) in human tumor cells [15,16,17], L-Met, D-Met, L-Ala, and D-Ala were evaluated.

## 2. Materials and Methods

[S-methyl-^3^H]-L-Met (^3^H-L-Met), [S-methyl-^3^H]-D-Met (^3^H-D-Met), [2, 3-^3^H]-L-Ala (^3^H-L-Ala), and [2, 3-^3^H]-D-Ala (^3^H-D-Ala) (American Radiolabeled Chemicals, St. Louis, MO, USA) were used in the in vitro studies. 2-Deoxy-2-[^18^F]fluoro-D-glucose (^18^F-FDG), which was synthesized at our PET facility, was used in the in vivo biological distribution study.

### 2.1. Bacterial Strain and Culture Conditions

*Escherichia coli* EC-14 (Shionogi, Osaka, Japan) was isolated as the clinical isolate strain. In pre-cultivation, EC-14 stock solutions with 50% glycerol were mixed with THY medium consisting of Todd–Hewitt Broth (Becton, Dickinson and Company, Franklin Lakes, NJ, USA) and 0.2% yeast extract (Becton, Dickinson and Company) in the ratio of 1:100 and incubated at 37 °C for 12–14 h with shaking. After incubation, EC-14 (1.2 × 10^8^ CFU/100 µL) was seeded in Dulbecco’s modified Eagle’s medium (DMEM; FUJIFILM Wako Pure Chemical Corporation, Osaka, Japan), which does not contain amino acids (amino-acid-free DMEM) and incubated at 37 °C with shaking, for the experiments. Bacterial protein was measured using the Pierce^TM^ BCA Protein Assay Kit (Thermo Fisher Scientific, Waltham, MA, USA).

### 2.2. Human Cell Line and Culture Conditions

The human keratinocyte cell line (HaCaT cells; Boukamp et al., 1988) [18] was used in accumulation experiments. As normal cells are difficult to culture, HaCaT cells established from adult male skin were used after considering technical issues in the research setup. HaCaT cells was cultured in DMEM containing 8% fetal bovine serum (FBS; Biosera, Kansas, MO, USA) and 100 U/mL penicillin/100 µg/mL streptomycin/250 ng/mL amphotericin B (FUJIFILM Wako Pure Chemical Corporation) at 37 °C in an atmosphere of <5% CO_2_.

### 2.3. Accumulation of ^3^H-L- and D-Met and Ala in E. coli EC-14

*E. coli* EC-14 at 1.2 × 10^8^ CFU/100 µL were seeded in 5 mL of amino-acid-free DMEM and incubated for 1, 2, 4, 6, 8, 12, and 24 h. After incubation, 37 kBq/10 µL of ^3^H-L-Met, ^3^H-D-Met, ^3^H-L-Ala, and ^3^H-D-Ala were added to the bacterial solution and incubated for 5, 30, and 60 min at 37 °C with gentle shaking. After conducting this time-course experiment in which the bacterial incubation time was varied, we investigated the accumulation of ^3^H-L-Met and ^3^H-D-Met in each of the *E. coli* EC-14 and HaCaT cells, for various incubation times after ^3^H-L-Met and ^3^H-D-Met were added. EC-14 was incubated in amino-acid-free DMEM for 4 h. EC-14 was then collected by centrifugation at 7000× g for 10 min at 4 °C, washed three times with 5 mL of phosphate-buffered saline (PBS; Medical & Biological Laboratories Co., Ltd., Aichi, Japan), and lysed by 1 mL of 0.1 M NaOH. Five hundred microliters of bacterial lysate was mixed with 5 mL of ULTIMA GOLD (Perkin Elmer, Waltham, MA, USA) and the radioactivity of the mixture was measured using a liquid scintillation counter (LSC-5100; Hitachi Aloka Medical, Tokyo, Japan). In measuring the incorporation rate of ^3^H-L- and D-Met into the protein fraction, another 190 µL of the bacterial lysate consisting of 0.1 M NaOH was mixed with 10 µL of 100 *w*/*v*% trichloroacetic acid (Nacalai Tesque, Kyoto, Japan). After mixing, the precipitated protein was collected on a glass fiber filter (GC-50; Advantec Toyo Kaisha, Tokyo, Japan), washed three times with 1 mL of ice-cold 5% trichloroacetic acid solution, and fixed by heating at 100 °C for 1 h. The fixed protein was mixed with 5 mL of ULTIMA GOLD and the radioactivity was measured and evaluated as the incorporation rate into the protein fraction.

### 2.4. Accumulation of ^3^H-L- and D-Met and Ala in HaCaT Cells

HaCaT cells were seeded at 5 × 10^4^ per cell in a 24-well plate in DMEM and cultured at 37 °C in an atmosphere of 5% CO_2_. For ^3^H-L-Met, ^3^H-D-Met, ^3^H-L-Ala, and ^3^H-D-Ala accumulation assay, DMEM was removed and 450 µL of amino-acid-free DMEM was added to the HaCaT cells followed by incubation at 37 °C for 10 min. We then added 37 kBq/50 µL of ^3^H-L-Met, ^3^H-D-Met, ^3^H-L-Ala, or ^3^H-D-Ala to the HaCaT cells, which were incubated at 37 °C for 5, 30, and 60 min with gentle shaking. After incubation, the cells were washed twice with 500 µL of ice-cold amino-acid-free DMEM and lysed by 1 mL of 0.1 M NaOH. Five hundred microliters of cell lysate was mixed with 5 mL of ULTIMA GOLD and the radioactivity of the mixture was measured by LSC-5100. The incorporation rates of each radiotracer into the protein fraction were measured the same as that described in the section “*Accumulation of ^3^H-L- and D-Met in E. coli EC-14”.*

### 2.5. E. coli EC-14 Lung-Infection-Model Mice

All applicable institutional guidelines of Kanazawa University for the care and use of animals were followed. All procedures were performed in accordance with the ethical standards of Kanazawa University (Animal Care Committee of Kanazawa University, AP-173851) and with international standards for animal welfare and institutional guidelines. EC-14 was pre-cultured in Luria–Bertani (LB) broth (Becton, Dickinson and Company) for 12 h followed by seeding into new LB broth in the ratio of EC-14 solution (LB broth = 1:100) and incubation at 37 °C for 12–14 h with shaking. After pre-incubation, EC-14 was collected by centrifugation at 8000× *g* for 5 min at 4 °C and suspended in amino-acid-free DMEM for inoculation to mice. Jcl:ICR mice (n = 3; male; age, 4 weeks; CLEA Japan, Tokyo, Japan) were purchased 7 days prior to infection and treated intraperitoneally with 150 mg/kg and 100 mg/kg of Endoxan (Shionogi) at 4 days and 1 day prior to infection, respectively. The lungs of mice were infected with EC-14 at 3–7 × 10^7^ colony forming units (CFU)/80 µL under isoflurane (FUJIFILM Wako Pure Chemical Corporation, Osaka, Japan) anesthesia. At 1, 2, 4, 8, and 24 h after infection, the infected mice were euthanized with isoflurane and the lung tissue was collected and homogenized with PBS. CFUs in homogenized solution were measured by serial dilution in PBS and plating on LB agar plates.

### 2.6. Biological Distribution of E. coli EC-14 Lung-Infection-Model Mice Using ^3^H-L-Met, ^3^H-D-Met, and ^18^F-FDG

EC-14 at 3–7 × 10^7^ CFU/80 µL was infected to the lung of immunosuppressed mice, as described in the “*E. coli EC-14 lung-infection-model mice*” section. At 4 and 24 h after infection, 50 kBq/50 µL of ^3^H-L-Met and ^3^H-D-Met, and 17 MBq/90 µL of ^18^F-FDG were administered intravenously to the mice (n = 4), which were fasted for 4–5 h prior to administration. Mice were euthanized at 15 and 60 min after administration and the blood, heart, lung, liver, spleen, pancreas, kidney, brain, bladder, infected lung, and contralateral noninfected lung were collected. Organ samples were adjusted to approximately 100 mg and mixed with 1 mL of ULTIMA GOLD, followed by the initial measurement of ^18^F-FDG radioactivity using a γ-counter (AccuFLEX ARC-8001; Hitachi Aloka Medical). The samples were incubated overnight at room temperature, mixed with 3 mL of ULTIMA GOLD and 100 µL of hydrogen peroxide, and then incubated for 1–2 more days. After incubation, the radioactivity of ^3^H-L-Met and ^3^H-D-Met was measured by LSC-5100.

Data are presented as means and standard deviation. *p* values were calculated using the two-tailed unpaired Mann–Whitney t-test for comparison between two groups using GraphPad Prism 8 statistical software (GraphPad Software, La Jolla, CA). A *p* value less than 0.01 or 0.05 was considered to indicate statistical significance. 

## 3. Results

### 3.1. Accumulation of ^3^H-L- and D-Met and -Ala in E. coli EC-14

The accumulation of ^3^H-L-Met, ^3^H-D-Met, ^3^H-L-Ala, and ^3^H-D-Ala in *E. coli* EC-14 at 5 min after each injection are shown in Figs 1 and 2. ^3^H-L-Met and ^3^H-D-Met accumulation was greatest at 4 h after incubation (Figure 1). The accumulation of ^3^H-L-Met and ^3^H-D-Met in the protein fraction was 4.4 and 2.4 nmol/g protein, respectively. ^3^H-L-Ala accumulation was greatest at 6 h after incubation (Figure 2). ^3^H-D-Ala accumulation was greatest after 12 h of incubation, in the stationary phase, but the amount of accumulation was little changed at 4 h or later after incubation. The accumulation of ^3^H-L-Ala and ^3^H-D-Ala in the protein fraction was 3.0 and 1.1 nmol/g protein, respectively.

### 3.2. Accumulation of ^3^H-L- and D-Met and -Ala in E. coli EC-14 and HaCaT Cells

The accumulation of ^3^H-L- and D-Met in *E. coli* EC-14 and HaCaT cells at 5 min, 30 min, and 60 min after injection is shown in Figure 3. The accumulation of ^3^H-L-Met in EC-14 at 5 min after incubation was 13.8 nmol/g protein including a protein fraction of 6.0 nmol/g protein (Figure 3a), whereas that in HaCaT cells was 1.71 nmol/g protein including a protein fraction of 0.19 nmol/g protein (Figure 3b). The accumulation of ^3^H-L-Met in EC-14 was stable over 60 min, whereas that in HaCaT cells increased to 15.9 nmol/g protein at 60 min after injection. In contrast, at 5 min after injection, ^3^H-D-Met showed 4.3 nmol/g protein including a protein fraction of 1.90 nmol/g protein, whereas that in HaCaT cells was 0.21 nmol/g protein including a protein fraction of 0.01 nmol/g protein. At 60 min after injection, the accumulation of ^3^H-D-Met in EC-14 increased to 10.3 nmol/g protein and the accumulation in HaCaT cells was 1.28 nmol/g protein.

The accumulation of ^3^H-L- and D-Ala in *E. coli* EC-14 and HaCaT cells at 5 min, 30 min, and 60 min after injection is shown in Figure 4. The accumulation of ^3^H-L-Ala in EC-14 at 5 min after incubation was 17.1 nmol/g protein, which was maintained over 60 min (Figure 4a), whereas that in HaCaT cells was 3.57 nmol/g protein at 5 min after injection, which increased to 13.0 nmol/g protein at 60 min (Figure 4b). In comparison, the accumulation of ^3^H-D-Ala in EC-14 was 9.87 nmol/g protein at 5 min after injection, which was maintained until 60 min; and that in HaCaT cells was 0.24 nmol/g protein at 5 min after injection and 1.70 nmol/g protein at 60 min after injection.

### 3.3. Growth Curve of E. coli EC-14 in Lung of Lung-Infection-Model Mice

The growth curve for *E. coli* EC-14 in lung-infection-model mice with *E. coli* EC-14 is shown in Figure 5. The average CFU of EC-14 was approximately 4.3 × 10^6^ at 1 h after infection, which increased to 1.4 × 10^7^ and 3.7 × 10^9^ at 4 and 24 h after infection, respectively. 

### 3.4. Biological Distribution of ^3^H-L-Met, ^3^H-D-Met, and ^18^F-FDG in EC-14 Lung-Infection-Model Mice

Table 1 shows the biological distribution of ^3^H-L-Met in EC-14 lung-infection-model mice. Compared with the noninfected control mice, the radioactivity in all harvested tissues except the heart, brain, and bladder was significantly greater at 4 h after infection, 15 min and 60 min after ^3^H-L-Met injection. The ratios of lung to heart were significantly higher in comparison with those of the controls at 4 h after infection, 60 min after ^3^H-L-Met injection; and also significantly higher in comparison with the controls at 24 h after infection, 15 min and 60 min after ^3^H-L-Met injection. The ratios of lung to liver were significantly higher in comparison with those of the controls at 4 h after infection, 15 min after ^3^H-L-Met injection; and were lower at 24 h after infection, 15 min and 60 min after ^3^H-L-Met injection. The ratios of infected lung to control lung were higher at 4 h after infection than at 24 h after infection.

Table 2 shows the biological distribution of ^3^H-D-Met in EC-14 lung-infection-model mice. Compared with the noninfected control mice, the radioactivity in all harvested tissues except the heart and bladder was significantly greater in all infected mice. The ratios of lung to heart were significantly higher in comparison with those of the controls at 24 h after infection, 15 min after ^3^H-D-Met injection. The ratios of lung to liver were significantly lower in comparison with those of the controls at 24 h after infection, 15 min and 60 min after ^3^H-D-Met injection. The ratios of infected lung to control lung were higher at 4 h after infection than at 24 h after infection.

Table 3 shows the biological distribution of ^18^F-FDG in EC-14 lung-infection-model mice. Compared with the noninfected control mice, the radioactivity in the lung, liver, spleen, and brain was significantly greater at 24 h after infection, 15 min and 60 min after ^18^F-FDG injection. The ratios of lung to heart were significantly higher in comparison with those of the controls at 24 h after infection, 15 min and 60 min after ^18^F-FDG injection. The ratios of lung to liver were significantly higher in comparison with those of the controls at 4 h and 24 h after infection and at 15 min after ^18^F-FDG injection. The ratios of infected lung to control lung were higher at 24 h after infection than at 4 h after infection.

## 4. Discussion

In this study, we conducted in vitro and in vivo studies to compare the L- and D-amino acid accumulation of bacterial infection. In the in vitro study, ^3^H-L-Met and ^3^H-D-Met showed the maximum accumulation in EC-14 at 4 h after incubation, in the early growth phase (Figure 1). ^3^H-L-Ala accumulation was greatest at 6 h after incubation, whereas ^3^H-D-Ala accumulation was greatest at 12 h after incubation, but there was little change in accumulation at 4 h or later after incubation (Figure 2). These findings suggest that L- and D-Met and Ala accumulation can indicate the growth activity of bacteria.

We compared the accumulation of ^3^H-L-Met, ^3^H-D-Met, ^3^H-L-Ala, and ^3^H-D-Ala in EC-14 at 4 h after incubation and in HaCaT cells with different incubation times after ^3^H-L-Met, ^3^H-D-Met, ^3^H-L-Ala, and ^3^H-D-Ala injection (Figs 3 and 4). The accumulation of ^3^H-L-Met and ^3^H-L-Ala was greater than that of ^3^H-D-Met and ^3^H-D-Ala in both EC-14 and HaCaT cells. In both D-amino acids, the accumulation was higher in EC-14 than in HaCaT cells. Avid accumulation was identified at 5 min after ^3^H-L- and D-Met and ^3^H-L- and D-Ala injection in EC-14, whereas the accumulation gradually increased in HaCaT cells over time. In EC-14, the proportion of accumulation of ^3^H-L-Met and ^3^H-L-Ala into the protein fraction was similar to that of ^3^H-D-Met and ^3^H-D-Ala. Human cells generally take in L-amino acids and then synthesize proteins to produce energy but take in few D-amino acids [19]. In contrast, the present results indicate that bacteria use both L-amino and D-amino acids. As D-amino acids are known to regulate the composition and strength of peptidoglycan layers and cell wall reinforcement [15,20,21], it may be possible that bacteria utilize both L-amino acids and D-amino acids as constituents. In addition, bacteria convert L-amino acids to D-amino acids and reverse in the bacterial body using the enzyme and amino acid racemase [22]. Therefore, D-amino acids may be converted to L-amino acids in bacteria, where they are used for protein synthesis. ^11^C-MeAIB, which is an analog of Ala, showed little accumulation in an in vitro study with EC-14 (no data), possibly because of the slight difference in its chemical structure compared with Ala. For this reason, we did not use ^3^H-L- or D-Ala in our biological distribution study. 

Muscle-infection-model mice have generally been used in bacterial infection imaging studies. However, we used lung-infection-model mice (Figure 5) for the reasons that sepsis- and organ-dysfunction-model mice are ideal for the bacterial infection imaging conducted in this study and because the lung infection model is highly applicable to clinical practice. The results of the study confirmed that the growth of EC-14 increased in the lungs of the mice with continuing time after infection.

Based on the results of the in vitro experiments, we conducted in vivo distribution experiments because we considered that the high accumulation contrast of labeled amino acids between *E. coli* and HaCaT cells could be applied to bacterial imaging. Using lung-infection-model mice, we conducted a biological distribution study in EC-14-infected-model mice at two injection time points, 4 h after infection as the early post-infection period and 24 h after infection as the late post-infection period (Table 1, Table 2 and Table 3). When we compared radiotracer accumulation in control mice and lung-infection-model mice at 4 h and 24 h after infection, at 15 min after ^3^H-L-Met injection, the radioactivity in all organs except the heart, brain, and bladder was significantly higher at 15 min after ^3^H-L-Met injection than at 60 min (Table 1). In the biological distribution study of ^3^H-D-Met, the radioactivity of all organs except the heart at 24 h after infection was significantly higher at 15 min after ^3^H-D-Met injection than at 60 min (Table 2). Although there is usually little accumulation of D-Met in the brain, accumulation in the brain was greater at 4 h and 24 h after infection compared with control mice. This finding suggests the spread of bacteria throughout the body. We confirmed bacteria of some colonies and approximately 1 × 10^5^ CFU in the blood at 4 h and 24 h, respectively, after infection of immunosuppressed mice. A previous study that introduced *E. coli* HB-101 into lung-infection-model mice in an immunosuppressed state following suppression of Lipocalin 2 (an effector molecule of the innate immune system) reported infection from the lung to other organs at 48 h after infection [23]. It can be assumed that bacteria in the lungs spread to the whole body via the bloodstream [24].

Overall, there was little difference in biological distribution between ^3^H-L-Met and ^3^H-D-Met except in the brain. Neumann et al. performed bacterial imaging using ^11^C-L-Met and D-Met in a murine myositis model at 12 h after infection of *E. coli* strain ATCC 25922 [12] and reported the superiority of ^11^C-D-Met compared with ^11^C-L-Met in imaging at about 60 min after each radiotracer injection. However, their use of a different strain of *E. coli* than that in the present study (EC-14) might have influenced their results, and the better contrast ratios of ^11^C-D-Met may have been due to the faster excretion of ^3^H-D-Met than ^3^H-L-Met in control mice (Table 1 and Table 2).

In the biological distribution of ^18^F-FDG, significant differences between the control mice and the mice at 4 h after infection were found in the lung and spleen at 15 min after ^18^F-FDG administration (Table 3). ^18^F-FDG accumulation is reported to indicate areas of inflammation after infection, as well as bacterial infection [25]. The ^18^F-FDG results of the present study suggested minimal areas of inflammation in the whole body because the accumulation of ^18^F-FDG at 4 h after infection was not significantly different to that in controls, except in the brain and spleen. Therefore, ^3^H-L- and D-Met are sensitive for the detection of areas of infection after the spread of bacteria throughout the body, whereas ^18^F-FDG can detect areas of primary infection.

It is difficult to evaluate the ratios of accumulation between the infected lung and other organs (heart and liver), which are commonly calculated in tumor imaging, using ^3^H-L- and D-Met because, in this mouse model, there was the whole-body spread of bacteria (Table 1 and Table 2); however, almost all of these ratios showed a significant increase compared with controls at 24 h after infection. Although these ratios were also significantly increased compared with controls at 24 h after infection when using ^18^F-FDG (Table 3), there was no significant difference in these ratios between the control and model mice at 4 h after infection, despite no significantly increased accumulation of ^18^F-FDG in the organs compared to the control. Therefore, ^3^H-L- and D-Met are more sensitive than ^18^F-FDG for detection of bacterial infection.

At 4 h after infection (early post-infection period), the ratios of accumulation in the infected lung/control lung were 2.80, 2.84, and 1.34 at 15 min after ^3^H-L-Met, ^3^H-D-Met, and ^18^F-FDG injection, respectively. At 24 h after infection (late post-infection period), these ratios were 2.11, 2.31, and 2.29; and for each radiotracer, the ratios at 60 min after injection were similar to those at 15 min after injection. In a clinical oncology study, ^18^F-FDG accumulation in non-small-cell lung cancer showed a more-than-two-fold increase compared with the background [26]. Thus, we can confirm that a two- to three-fold accumulation ratio in the infected lung/control lung in this study is sufficient for clinical application. Therefore, ^3^H-L- and D-Met showed comparable performance in in vivo experiments in EC-14 lung-infection-model mice, and they could be superior to ^18^F-FDG in terms of detecting bacterial infection at the early stage of infection.

As a limitation of this study, we could not perform ^11^C-Met imaging, for the reason that our facility is not equipped with a cyclotron or an automatic synthesizer for ^11^C labeling. However, the results of the whole-body biodistribution are considered equivalent to those of whole-body imaging.

## 5. Conclusions

^11^C-L-Met and D-Met, which differ from ^3^H-L-Met and D-Met only in terms of their radioisotopic labeling, showed comparable performance in in vivo experiments in EC-14 lung-infection-model mice, and are more sensitive than ^18^F-FDG for detecting bacterial infection at the early stage of infection.

## Figures and Tables

**Figure 1 ijms-23-02467-f001:**
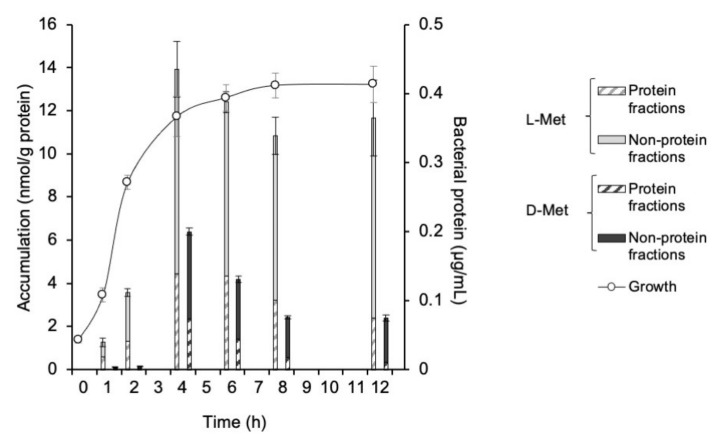
Accumulation of ^3^H-L-Met and ^3^H-D-Met in *E. coli* EC-14 at 5 min after each injection. EC-14 was incubated in amino-acid-free DMEM for 1, 2, 4, 6, 8, and 12 h. The accumulation of ^3^H-L-Met and ^3^H-D-Met was greatest at 4 h after incubation. The incorporation rate of ^3^H-L-Met was higher than that of ^3^H-D-Met at all EC-14 incubation times.

**Figure 2 ijms-23-02467-f002:**
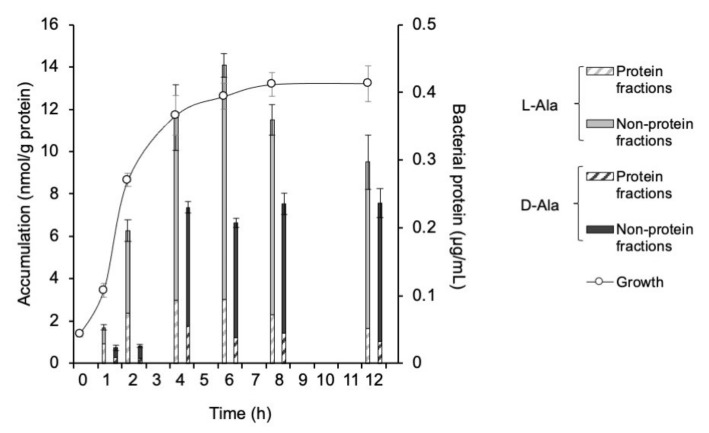
Accumulation of ^3^H-L-Ala and ^3^H-D-Ala in *E. coli* EC-14 at 5 min after each injection. EC-14 was incubated in amino-acid-free DMEM for 1, 2, 4, 6, 8, and 12 h. The accumulation of ^3^H-L-Ala was greatest at 6 h after incubation, whereas the accumulation of ^3^H-D-Ala was greatest after an incubation time of 12 h; however, there was little change in accumulation at 4 h or later after incubation. The incorporation rate of ^3^H-L-Ala was higher than that of ^3^H-D-Ala at all EC-14 incubation times.

**Figure 3 ijms-23-02467-f003:**
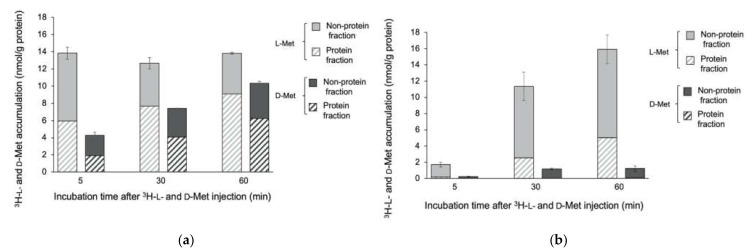
Accumulation of ^3^H-L-Met and ^3^H-D-Met in (**a**) *E. coli* EC-14 at 4 h after incubation and in (**b**) HaCaT cells at 5, 30, and 60 min after ^3^H-L-Met and ^3^H-D-Met injection. The accumulation of ^3^H-L-Met and ^3^H-D-Met was greater in EC-14 at 5 min and later after incubation than in HaCaT cells. Similar incorporation rates of ^3^H-L-Met and ^3^H-D-Met were identified in accumulation into EC-14.

**Figure 4 ijms-23-02467-f004:**
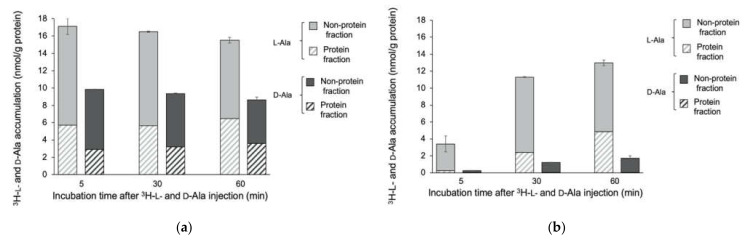
Accumulation of ^3^H-L-Ala and ^3^H-D-Ala in (**a**) *E. coli* EC-14 at 4 h after incubation and (**b**) accumulation of ^3^H-L-Ala and ^3^H-D-Ala in HaCaT cells at 5, 30, and 60 min after ^3^H-L-Met and ^3^H-D-Met injection. The accumulation of ^3^H-L-Ala and ^3^H-D-Ala in EC-14 at 5 min or later after incubation was higher than that in HaCaT cells. Similar incorporation rates of ^3^H-L-Ala and ^3^H-D-Ala were identified in accumulation into EC-14.

**Figure 5 ijms-23-02467-f005:**
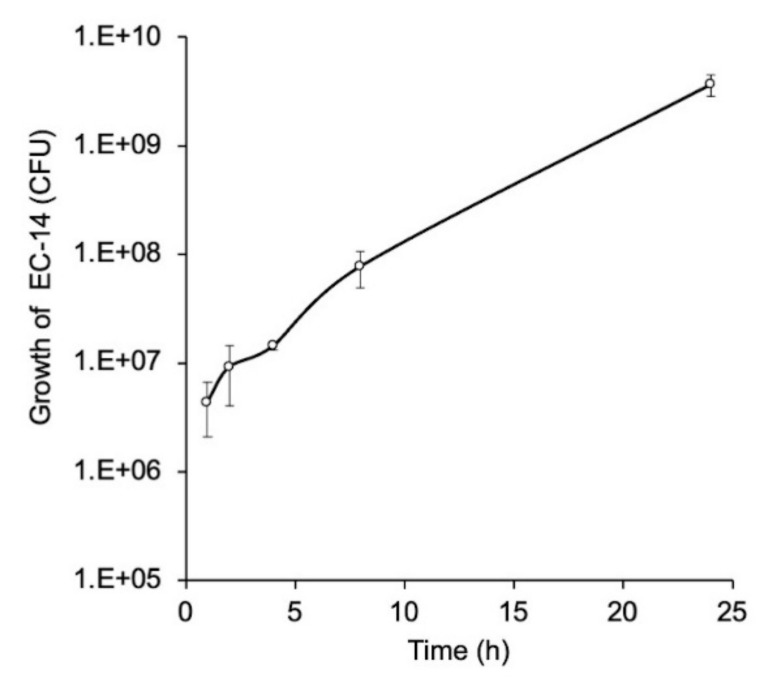
Growth curve of *E. coli* EC-14 in lung-infection-model mice. EC-14 at 3–7 × 10^7^ CFU/80 µL was infected to the lung of mice. At 1, 2, 4, 8, and 24 h after infection, the mean CFU of EC-14 was approximately 4.3 × 10^6^, 9.2 × 10^6^, 1.4 × 10^7^, 7.8 × 10^7^, and 3.7 × 10^9^, respectively.

**Table 1 ijms-23-02467-t001:** Biological distribution of ^3^H-L-Met in lung-infection-model mice.

Time after ^3^H-L-Met Injection	15 min	60 min	15 min	60 min	15 min	60 min
Organ (%ID/g)	Control		4 h after Infection		24 h after Infection	
Blood	0.50 ± 0.18	0.50 ± 0.43	1.61 ± 0.82 *	1.51±0.07 ^†^	2.74 ± 0.86 ^†^	1.24 ± 0.86 *
Heart	1.65 ± 0.74	1.89 ± 0.31	2.96 ± 0.96	3.74± 0.10 ^†^	1.35 ± 0.19	1.40 ± 0.27
Lung	1.49 ± 0.19	1.57 ± 0.26	4.16 ± 0.60 ^†^	4.06 ± 0.12 ^†^	3.13 ± 1.09 *	2.47 ± 0.10 ^†^
Liver	1.66 ± 0.53	2.09 ± 0.84	3.16 ± 0.20 ^†^	4.74 ± 0.14 ^†^	6.19 ± 0.93 ^†^	9.43 ± 0.82 ^†^
Spleen	1.39 ± 0.57	2.47 ± 0.63	7.06 ± 1.81 ^†^	6.09 ± 2.79 *	3.01 ± 0.55 ^†^	3.23 ± 1.81
Pancreas	4.29 ± 4.68	12.47 ± 4.68	22.01 ± 6.38 ^†^	21.69 ± 3.40 *	14.39 ± 2.53 ^†^	11.94 ± 0.94
Kidney	1.67 ± 0.62	3.17 ± 0.49	5.13 ± 2.83 *	5.92± 0.73 ^†^	5.15 ± 0.40 ^†^	4.43 ± 0.72 *
Brain	4.10 ± 4.61	0.51 ± 0.10	2.80 ± 3.39	1.11 ± 0.48	1.41 ± 0.17	1.36 ± 0.18 ^†^
Bladder	0.20 ± 0.17	0.35 ± 0.27	0.55 ± 0.36	1.15± 0.42 *	0.46 ± 0.41	0.46 ± 0.41 *
Lung/heart	1.03 ± 0.39	0.83 ± 0.03	1.51 ± 0.51	1.09 ± 0.02 ^†^	2.27 ± 0.47 ^†^	1.82 ± 0.41 ^†^
Lung/liver	0.94 ± 0.20	0.85 ± 0.38	1.32 ± 0.20 *	0.86 ± 0.03	0.50 ± 0.13 *	0.26 ± 0.04 *
Infected lung /control lung	–	–	2.80	2.58	2.11	1.57

%ID/g indicates percent injected dose per gram of tissue. ^†^ *p* < 0.01 and * *p* < 0.05 compared with control at 15 min or 60 min after injection of ^3^H-L-Met.

**Table 2 ijms-23-02467-t002:** Biological distribution of ^3^H-D-Met in lung-infection-model mice.

Time after ^3^H-D-Met Injection	15 min	60 min	15 min	60 min	15 min	60 min
Organ (%ID/g)	Control		4 h after Infection		24 h after Infection	
Blood	0.34 ± 0.14	0.27 ± 0.07	1.71 ± 0.17 ^†^	1.09 ± 0.08 ^†^	2.54 ± 0.83 ^†^	2.27 ± 0.66 ^†^
Heart	1.01 ± 0.25	0.92 ± 0.31	2.32 ± 0.07 ^†^	2.18 ± 0.38 ^†^	1.25 ± 0.09	1.28 ± 0.30
Lung	1.01 ± 0.10	1.02 ± 0.33	2.87 ± 0.28 ^†^	2.52 ± 0.55 ^†^	2.34 ± 0.09 ^†^	2.34 ± 0.62 ^†^
Liver	1.14 ± 0.13	1.64 ± 0.80	3.01 ± 0.26 ^†^	3.80 ± 0.26 ^†^	4.32 ± 0.24 ^†^	7.12 ± 0.75 ^†^
Spleen	1.29 ± 0.25	2.01 ± 0.30	4.70 ± 0.37 ^†^	6.78 ± 2.16 ^†^	2.06 ± 0.42 *	4.05 ± 1.12 *
Pancreas	7.70 ± 1.25	8.77 ± 3.45	19.14 ± 2.18 ^†^	16.56 ± 4.03 *	12.07 ± 1.57 ^†^	15.68 ± 3.51 *
Kidney	3.36 ± 0.89	3.10 ± 0.42	8.79 ± 0.67 ^†^	8.15 ± 1.26 ^†^	9.11 ± 1.55 ^†^	10.25 ± 1.43 ^†^
Brain	0.44 ± 0.06	0.59 ± 0.13	1.21 ± 0.17 ^†^	1.28 ± 0.36 *	0.92 ± 0.15 ^†^	1.60 ± 0.15 ^†^
Bladder	0.62 ± 0.36	0.70 ± 0.49	1.94 ± 0.41 ^†^	0.69 ± 0.13	2.01 ± 0.85 *	–
Lung/heart	1.06 ± 0.33	1.20 ± 0.50	1.24 ± 0.14	1.17 ± 0.23	1.88 ± 0.18 ^†^	1.88 ± 0.52
Lung/liver	0.91 ± 0.20	0.68 ± 0.24	0.96 ± 0.10	0.67 ± 0.18	0.54 ± 0.03 *	0.33 ± 0.10 *
Infected lung /control lung	–	–	2.84	2.48	2.31	2.30

%ID/g indicates percent injected dose per gram of tissue. ^†^ *p* < 0.01 and * *p* < 0.05 compared with control at 15 min or 60 min after injection of ^3^H-D-Met/ The bladders of some mice could not be collected.

**Table 3 ijms-23-02467-t003:** Biological distribution of ^18^F-FDG in lung-infection-model mice.

Time after ^18^F-FDG Injection	15 min	60 min	15 min	60 min	15 min	60 min
Organ (%ID/g)	Control		4 h after Infection		24 h after Infection	
Blood	1.98 ± 0.58	0.45 ± 0.09	1.84 ± 0.85	0.41 ± 0.06	2.59 ± 0.72	0.49 ± 0.15
Heart	13.14 ± 7.34	11.23 ± 6.51	18.28 ± 8.61	17.07 ± 7.99	5.72 ± 3.23 *	9.85 ± 5.13
Lung	3.41 ± 0.34	3.57 ± 0.80	4.58 ± 0.91 ^†^	3.99 ± 0.71	7.82 ± 1.21 ^†^	9.31 ± 1.79 ^†^
Liver	2.85 ± 1.39	0.99 ± 0.21	2.47 ± 0.54	0.89 ± 0.13	4.65 ± 0.94 ^†^	2.34 ± 1.04 ^†^
Spleen	2.68 ± 0.68	3.42 ± 0.47	3.44 ± 0.63 *	3.53 ± 1.24	5.92 ± 2.90 ^†^	9.86 ± 6.01 ^†^
Pancreas	2.42 ± 1.57	1.69 ± 0.31	1.64 ± 0.30	1.61 ± 0.11	2.66 ± 1.79	2.16 ± 0.35 *
Kidney	5.97 ± 3.73	1.90 ± 0.51	5.40 ± 2.77	3.30 ± 1.82	6.69 ± 0.74	3.14 ± 0.54 ^†^
Brain	5.12 ± 2.35	6.39 ± 1.03	6.97 ± 1.46	5.11 ± 1.49	12.46 ± 1.53 ^†^	11.51 ± 2.78 ^†^
Bladder	1.50 ± 0.84	2.04 ± 1.19	1.86 ± 0.87	1.76 ± 1.13	1.67 ± 1.13	-
Lung/heart	0.36 ± 0.22	0.43 ± 0.33	0.36 ± 0.34	0.30 ± 0.18	1.72 ± 0.76 ^†^	1.15 ± 0.52 ^†^
Lung/liver	1.34 ± 0.37	3.86 ± 1.60	1.88 ± 0.31 ^†^	4.52 ± 0.84	1.71 ± 0.23 *	4.49 ± 1.49
Infected lung /control lung	-	-	1.34	1.12	2.29	2.61

%ID/g indicates percent injected dose per gram of tissue. ^†^ *p* < 0.01 and * *p* < 0.05 compared with control at 15 min or 60 min after injection of ^18^F-FDG. The bladders of some mice could not be collected.

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
