# Peer review of "Comparison of L- and D-Amino Acids for Bacterial Imaging in Lung Infection Mouse Model"

_ijms, 2022, doi:10.3390/ijms23052467_

Round 1

Reviewer 1 Report

In this work, the authors have used new biomarkers (3H-L-Met, 3L-D-Met, 3H-D-Ala and 3-H-D-Ala) to detect bacterial infection in the murine model of infection using bacterial imaging. The authors have enlarged the field of biomarkers that can used to image and study bacterial infection. The authors observed that L-enantiomer of both Alanine and Methionine were incorporated rapidly in EC-14 and HaCaT compared to their L-enantiomer.  3H-L-Met, 3H-L-D-Met show maximum accumulation in EC-14 after 4hrs whereas 3H-D-Ala the slowest accumulation attaining its maximum accumulation after 12h whereas 3H-D-Ala showed maximum accumulation after 6h. The authors also found that the HaCaT preferentially uses the L-enantiomer of both Met and Ala whereas the bacteria use both the forms of enantiomer. The authors have used these biomarkers to image the accumulation of these enantiomers in different organs in the murine model of infection. The authors found that 4hrs or 24hrs after bacterial infections and only after 15min after addition of the biomarker, the authors were able to detect the biomarker at all the organs except heart. This work is clearly written, the materials and methods are clearly explained and should be of interest to broader audience involved in the studying of bacterial pathogenicity.

Author Response

Dear reviewer 1

Thank you for your review. 

We are glad to get your good feedback and comments. 

Just in case, we will send you our revised file based on the comments of reviewer 2.

Best regards,

Yuka Muranaka

Reviewer 2 Report

In this manuscript the authors compared radiolabelled L- and D-amino acids for detecting early stage of infection. Biological distribution was assessed in control and lung infection model mice. There was little difference in biodistribution between 3H-L-and D-Met except in the brain. 3H-L- and D-Met were sensitive for detecting areas of infection after spread of bacteria throughout the body, whereas 18F-FDG mainly detected primary infection areas.

Overall the manuscript is well-written and provides useful information.

The methodology used is adequate and the results are well-described.

Minor comment:

Abstract and main text: "may be superior to 18F-FDG" could be changed in "could be superior to 18F-FDG".

Author Response

Dear reviewer 2,

Thank you for your comments. We appreciate it.

We are glad to get your good feedback and comments.

Our manuscript has been revised as the attached file based on your comments.

Please make sure it.

Best regards,

Yuka Muranaka
